# Breeding Tomato Hybrids for Flavour: Comparison of GWAS Results Obtained on Lines and F1 Hybrids

**DOI:** 10.3390/genes12091443

**Published:** 2021-09-18

**Authors:** Estelle Bineau, José Luis Rambla, Santiago Priego-Cubero, Alexandre Hereil, Frédérique Bitton, Clémence Plissonneau, Antonio Granell, Mathilde Causse

**Affiliations:** 1INRAE, UR1052, Genetics and Breeding of Fruit and Vegetables, 67 Allée des Chênes, Centre de Recherche PACA, Domaine Saint Maurice, CS60094, 84143 Montfavet, France; estelle.bineau@inrae.fr (E.B.); alexandre.hereil@inrae.fr (A.H.); frederique.bitton@inrae.fr (F.B.); 2GAUTIER Semences, Route d’Avignon, 13630 Eyragues, France; clemence.plissonneau@gautiersemences.com; 3Instituto de Biología Molecular y Celular de Plantas, CSIC, Universidad Politécnica de Valencia, 46022 Valencia, Spain; jrambla@alumni.upv.es (J.L.R.); S.Priego@biologie.uni-muenchen.de (S.P.-C.); agranell@ibmcp.upv.es (A.G.); 4Departamento de Ciencias Agrarias y del Medio Natural, Universitat Jaume I, 12071 Castellón de la Plana, Spain

**Keywords:** tomato, breeding, flavour, volatiles, GWAS

## Abstract

Tomato flavour is an important goal for breeders. Volatile organic compounds (VOCs) are major determinants of tomato flavour. Although most tomato varieties for fresh market are F1 hybrids, most studies on the genetic control of flavour-related traits are performed on lines. We quantified 46 VOCs in a panel of 121 small fruited lines and in a test cross panel of 165 hybrids (the previous panel plus 44 elite cherry tomato lines crossed with a common line). High and consistent heritabilities were assessed for most VOCs in the two panels, and 65% of VOC contents were strongly correlated between lines and hybrids. Additivity was observed for most VOCs. We performed genome wide association studies (GWAS) on the two panels separately, along with a third GWAS on the test cross subset carrying only F1 hybrids corresponding to the line panel. We identified 205, 183 and 138 associations, respectively. We identified numerous overlapping associations for VOCs belonging to the same metabolic pathway within each panel; we focused on seven chromosome regions with clusters of associations simultaneously involved in several key VOCs for tomato aroma. The study highlighted the benefit of testcross panels to create tasty F1 hybrid varieties.

## 1. Introduction

Tomato is the vegetable crop that is most consumed and produced worldwide [1]. Nevertheless, tomato flavour has been widely criticized [2,3]. Organoleptic quality can be summarized through flavour and texture. Flavour is a complex characteristic related to the integrated perception of taste (sugars and acids) and aroma through olfaction and retronasal olfaction of volatile organic compounds (VOCs) [4]. Up to now, more than 400 VOCs have been identified in tomato fruits [5]. These compounds belong to several metabolic pathways [6,7] and derive from: benzenoids (B-der); branched chain amino acids (BCAA-der); carotenoids (C-der); lipids (L-der); phenylalanine (Phe-der); sulphur containing compounds (S-der); and terpenoids (T-der). A subset of approximately 30 VOCs (listed in Table 1) has been suggested to impact tomato flavour, based on their odour threshold, their abundancy or their correlation to consumer liking [5,8]. The flaws of abundancy and odour unit as indicators of VOC contribution to flavour have been underlined in several studies. For example, although (Z)-3-hexenal is the most abundant volatile in tomato fruit, it does not correlate with consumer liking [8]. Consumer sensitivity to VOCs depends on the receptors they possess, with high allelic variability in the sequence of olfactive receptor genes leading to variable odour perception [9]. Moreover, cross-talk exists between taste and VOC perception, making it possible to enhance sweetness perception by increasing 2-phenylethanol content, for example [10,11]. Thus, flavour is a combination of human individual specificities, and misunderstood interactions between olfactory and taste receptors. VOC release also depends on their interaction with the tomato matrix [12], this characteristic being modulated by the ripening stage and the plant genotype [13]. In addition to being expensive, VOC quantification is technically demanding and requires both gas chromatography performance and a broad database to link VOC identity to the chemical structure and retention time measured with the chromatogram. Such a challenge is illustrated with the terpenoid-derived volatiles that are rarely quantified, although they show potential for tomato aroma improvement and are crucial to some white grape species and most Citrus species sensory quality [6,8].

Organoleptic quality-evaluated through both flavour and texture-depends on the variety choice, the growth conditions and post-harvest storage conditions [17]. However, our major interest lies in the genetic aspect of flavour-related traits and more specifically in VOC synthesis in fruits. The highly polygenic nature of VOC content has been revealed through multiple studies. The genetic control of 18 VOCs and sensory traits was first investigated by [17] in a Recombinant Inbred Lines (RILs) population. They highlighted several chromosome regions aimed at providing breeders with efficient tools for Marker Assisted Selection (MAS). Introgression Lines (ILs) derived from crosses between domesticated tomato (*Solanum lycopersicum*) and the wild species *Solanum pennellii* concomitantly enabled the cloning of major genes regulating key VOC synthesis [18,19]. Other tomato wild relatives such as *Solanum Pimpinellifolium* or *Solanum Habrochaites* have also been studied through biparental segregating populations and provided new QTLs for flavour-related traits as reviewed in [20]. Precise QTL mapping allowed by genome-wide association studies (GWAS) later brought better resolution around QTLs, with the dissection of previously identified large genomic regions into many single QTLs, which provided easier identification of candidate genes (‘CGs’) [21,22,23].

Although tremendous improvement has been achieved for QTL identification on homozygous plant material [20], breeders still need insight into the flavour improvement they can expect from these QTLs in their F1 hybrids, as no study has yet been undertaken on the evolution of flavour-related traits within heterozygous plant material. In the European Union, 89% of registered tomato varieties were hybrids in 2018 [24]. Frequent breeding programs include an ‘agronomic’ line that accumulates desirable disease resistance genes and agronomical value, and a line with good organoleptic quality [25,26]. We thus need to characterize the impact of the heterozygous state on flavour-related traits, and to identify the loci where heterozygosity (resp. homozygosity) is preferable to homozygosity (resp. heterozygosity) for flavour improvement. Similarly, little is known about the mode of inheritance of flavour-related traits and of their associated QTLs. The difficulty to bring forth aromatic hybrids must arise from unexpected modes of inheritance, as shown in [27] with over-recessive to over-dominant mode of inheritance for QTLs found for reproductive traits, but also for sugar content. With increasing consumer expectations for flavour in commercialized varieties, and tomato breeders requiring insight into the most promising QTLs to use in F1 hybrid tomato plants, we herein suggested a strategy to achieve flavour improvement.

We thus studied the genetic diversity within small fruited cultivated tomatoes for flavour-related traits in three related GWAS panels: one panel grown in 2018 composed of 121 cherry tomato (SLC) and wild relative *S. pimpinellifolium* (SP) homozygous lines; a second panel grown in 2019 composed of a test cross gathering the previous panel completed with 44 elite cherry tomato, all crossed by a common big fruited *S. lycopersicum* (SL) parent; a third panel derived from the test cross, carrying only the 121 hybrids from the core collection to compare VOC inheritance at the line and hybrid levels (Figure 1). Contrary to SL which underwent improvement sweeps with extreme reduction of its genetic diversity [28] and strong population structure preventing good resolution around QTLs [29], SP and SLC still harbour the genetic, flavour and metabolic diversity necessary for flavour improvement [21,30] without causing as much linkage drag as other wild relatives when used in breeding programs. We aimed at identifying consistent associations across lines and F1 hybrid panels or F1 hybrid specific associations. Two main issues are thus addressed: (i) identifying genomic regions harbouring potential for aroma improvement to facilitate both cherry type and big fruited tomato breeding for aroma; and (ii) suggesting novel candidate genes (‘CGs’) in promising F1 hybrid associations to improve our understanding of the genetic control of VOC accumulation in ripe tomato fruits.

## 2. Materials and Methods

### 2.1. Plant Material

We explored the phenotypic diversity of three related GWAS panels representative of the genetic diversity of small fruited tomatoes (Figure 1). The first panel was composed of 121 homozygous lines (106 SLC and 15 SP) from the core collection built by the French National Research Institute for Agriculture, Food and Environment (INRAE) of Avignon, France as described in [31]. We will further refer to this first panel as CCI (Core Collection INRAE). The second panel was composed of 165 F1 hybrids all sharing the common big fruited line Ferum TMV (FTMV) as tester. We crossed FTMV with (i) CCI and (ii) 44 elite cherry tomato lines created by the breeding company Gautier semences. F1 hybrid ID starting with an “X” originated from the Gautier semences plant material. This F1 hybrid GWAS panel will be further referred to as TCT (Test Cross Total). To compare GWAS results from similar panels studied at the line or F1 hybrid level, we extracted from the TCT dataset the subset TCI (Test Cross INRAE), which consisted of the 121 F1 hybrids derived from the cross between CCI and FTMV. We performed a third GWAS on this panel.

### 2.2. Growth Conditions

The CCI panel was grown from March to July 2018 under passive glasshouse irrigated conditions–in soil-on the experimental site of INRAE GAFL, Avignon, France. The TCT-which contained the subset TCI-panel was grown from April to July 2019 under soilless and passive irrigation conditions in a plastic greenhouse on the experimental site of the seed company Gautier semences, Eyragues, France. At least three plants per genotype were cultivated in both trials.

### 2.3. Fruit Samples

Three harvests—representing three replicates—of red ripe fruits were conducted for each trial during three successive weeks, starting from the second truss for the first harvest and finishing around the fifth truss for the third harvest. At least 10 fruits were harvested from each plot and each harvest (up to 30 fruits per harvest for SP accessions). The harvested fruits were divided in two pools: the pericarp from at least 5 fruits per plot was fast frozen in liquid nitrogen and ground to powder with a cryogenic mill. The samples were then stored at −80 °C until analysis of their VOC content. The second pool of fruits was weighted to get an average fruit weight. The fruits were then crushed and the resulting juice was used for Soluble Solid Content (SSC) measurement. Phenotypic values from the CCI and TCI panels are available in Appendix A.

### 2.4. VOC Profiling by Gas Chromatography/Mass Spectrometry

Profiling of volatile compounds was performed at the Instituto de Biología Molecular y Celular de Plantas, at the Universidad Politécnica de Valencia, Spain. The same protocol was used in 2018 on CCI and in 2019 on TCT. Right before analysis, 0.5 g of each sample was weighed in a 15 mL vial. The samples were then incubated at 37 °C for 10 min. Then, 1.1 g of CaCl_2_.2H2O and 500 µL of a 100 mM EDTA-NaOH pH 7.5 solution were added, gently mixed and sonicated for 5 min. Finally, 1 mL of the resulting paste was transferred to a 10 mL screw cap vial for analysis. Extraction of VOCs was performed by means of headspace solid phase microextraction (HS-SPME) due to its high sensitivity [32]. Briefly, vials were first pre-incubated at 50 °C with 500 rpm agitation for 10 min, followed by 20 min extraction with a 65 µm PDMS/DVB SPME fibre (SUPELCO) under the same conditions. Extracted VOCs were injected in a 6890 N gas chromatograph coupled to a 5975B mass spectrometer (Agilent Technologies), and compounds desorbed at 250 °C for 1 min in splitless mode. Chromatographic conditions were: 40 °C for 2 min; 5 °C/min ramp up to 260 °C; 260 °C for 5 min; 1.2 mL constant helium flow; DB-5ms capillary column (60 m, 0.25 mm, 1 µm). Detection was performed in the scan mode in the m/z range 35–250 (6.2 scans/s), with EI ionization at 70 eV. The amount of 84 targeted VOCs was recorded with the Enhanced Data Analysis (Agilent) software. A homogenate comprising all the tomato samples evaluated in the trial was injected on a daily basis and used as a reference for correction of instrumental drift and fibre aging. The amount of each volatile in a given sample was expressed as the relative amount of the volatile compared to its abundancy detected in the tomato homogenate comprising all the samples evaluated in the trial.

### 2.5. Data Processing and Statistics

The same statistical analyses were performed on the three panels separately using the R software v.3.6.2 [33]. Prior to GWAS analyses, a fixed effect analysis of variance was conducted with the *car* package on each panel to test for genotype effect with the following model:y_ij_ = μ + g_i_ + r_j_ + ε_ij_,(1)
where y_ij_ is the trait value of genotype I in harvest j, μ is the intercept, g_i_ and r_j_ represent the fixed effects of the genotype and the harvest, respectively, and ε_ij_ the residual effect. Except for three terpenoid-derived VOCs in the CCI panel and one lipid-derived VOC in the TCT panel, all VOCs, SSC and fruit weight showed significant genotype effects. Broad-sense heritability (h^2^) was computed for every trait with the *lme4* package by using the following linear mixed model:y_ij_ = μ + g_i_ + r_j_ + ε_ij_,(2)
where y_ij_ is the trait value of genotype i in harvest j, μ is the intercept, g_i_ the random effect of genotype i, r_j_ the fixed effect of harvest j, and ε_ij_ the random residual effect. Then, heritability was derived from the variance components of the model as:h^2^ = σ_G_^2^/(σ_G_^2^ + σ_e_^2^),(3)
where σ_G_^2^ and σ_e_^2^ are the genetic and residual variance, respectively. Skewed data were then transformed using $\sqrt{x}$, $\frac{1}{\sqrt{x}}$ or $\frac{1}{x}$ transformations to meet normality assumptions required for GWAS analysis.

Correlations between traits within and among panels were computed using Pearson’s pairwise correlations on scaled datasets. The graphic representation of the pairwise correlations were produced using the R packages *corrplot* and the ‘hclust’ clustering method. Correlations between identical traits between CCI and TCI were computed on scaled datasets with Pearson’s pairwise correlations.

Metabolomic profiles from each accession were produced with the R packages *ComplexHeatmap* and *dendextend* on scaled datasets. Clusters for metabolites and accession profiles were produced with the R package *hclust* and the ‘average’ method on scaled dataset. The *factoextra* package evaluated the optimal number of clusters when a bend was observed in the total within sum of square as the number of clusters increased. Metabolic clusters were further validated with the *pvclust* R package based on ‘euclidian’ distance, ‘average method, and *n* = 1000 bootstrap replications. Briefly, *pvclust* calculated probability values for each cluster using bootstrap resampling techniques. *p*-values are given as ‘approximately unbiased’ (AU) or ‘bootstrap probability’ (BP), both reflecting the frequency of the cluster appearance across the multiple resampling (Suzuki and Shimodaira 2006). Metabolic clusters gathering compounds belonging to the same metabolic pathway allowed a quality control of the VOC quantification method. PLS-DAs were computed on scaled data with the R package *mixOmics*. To assess the statistical link between VOCs, SSC and each genetic group, we computed *p*-values with the ‘catdes’ function of the *FactomineR* package in order to support metabolic signatures identified by the PLS-DA method.

### 2.6. GWAS Analysis

The core collection, FTMV and the Gautier semences plant materials were genotyped and filtered on quality modalities for 7442 common SNP markers (Appendix A), using the Infinium assay developed by the Solanaceae Coordinated Agricultural Project (‘SolCAP’) [34,35]. Each genotypic matrix was first filtered with the *plink* software [36] with: MAF ≤4%, missing genotype per accession ≤20% and missing information per marker ≤15%. As a result, genotypic matrices counted 5996 identical markers for CCI and TCI, and 6058 SNP markers for TCT. We conducted the GWAS using the multi-locus mixed model (‘MLMM’) proposed by [37]. Structure and Kinship matrices were re-evaluated for each trait in the GWAS analyses with the R package *FactomineR* and the function ‘emma.kinship’ (Identity By State method), respectively [38,39]. The detection threshold was set at pval <10^−4^ with up to 15 cofactors allowed. For any significant association, we first computed the marker pairwise linkage disequilibrium (‘LD’) with each neighbour marker in a 2 Mbp interval (1 Mb on each chromosome end) with the *LDcorSV* package [40]. This package took both structure and kinship matrices into account to compute local r^2^ between pairs of markers on a given chromosome. Thus, the confidence interval (‘CI’) around each associated SNP was defined as the interval where the computed r^2^ was higher than 0.5 in a 2 Mbp region. The SL4.0 tomato genome version was used for the analysis along with the annotation ITAG4.0. We excluded genes that did not map on any of the 12 chromosomes, and focused on the 33,562 remaining genes. From these CIs, we extracted the list of CGs comprised in each interval. We retrieved CGs when their functional annotations matched functions involved in VOC synthesis as reported in the bibliography, irrespective of their expression in tomato fruits. Lists of CGs per metabolic pathway were thus constituted (Appendix A) and then matched to the genes found in each association CI. We also assessed the number of transcription factors (‘TFs’) in each CI, based on the list of TFs presented in Appendix A. We reported genes expressed in fruits based on transcriptome analysis of fruits from the eight parental lines of the MAGIC population described in [41]. Two SLC they studied are included in our panels at the line (CCI) and hybrid (TCI and TCT) levels.

We performed a χ^2^ test on each metabolic pathway and each panel separately to assess the ability of the MLMM method combined with the CI definition to detect CGs. We compared the number of CGs identified in the CI of associations for a specific metabolic pathway and a given panel, to the number of CGs in the whole genome for a specific metabolic pathway (Appendix A). The χ^2^ test was computed with a 5% error threshold.

### 2.7. QTL Compilation

Associations from the two GWAS panels were first compared in order to identify association consistency or specificity across panels. Overlapping or specific associations were analysed in regard to the trait metabolic pathway. We defined overlaps as associations for traits belonging to the same metabolic pathway, and which CIs partially overlapped, or were separated by less than 0.1 Mpb. CGs identified in the CCI, TCI and TCT associations were then further compared with those suggested in a review from [20] which encompassed QTLs reported for aroma from 16 studies performed on GWAS panels or biparental segregating populations.

### 2.8. Putative Effect of Polymorphisms in Candidate Genes

For the subset of CGs we selected, we looked for the putative effect of polymorphisms that could be found when we compared sequencing data from 47 lines from the CCI panel against the reference genome Solanum lycopersicum Heinz_1706. These lines appear in bold type in Appendix A. We downloaded all sequences from the NCBI platform with the SRA Toolkit v2.10.0 and examined quality with *FastQC* V0.11.8 [42] and *MultiQC* v1.7 [43]. Sequences were then trimmed with *fastp* 0.20.0 [44] with these parameters: max_len1 350, cut_mean_quality 20, cut_window_size 4, complexity_threshold 30. For each accession, fastq files from several libraries were merged when several were available, and were aligned on the reference genome Heinz_1706 v.4.0.0 (available on https://solgenomics.net, accessed on 12 September 2021) using *bwa* 0.7.17 and PCR duplicates were removed with *SAMtools* v1.9 [45] sambamba v0.7.1. Variants were called for each accession with *gatk4* v4.1.4.1 [46] and a genomic database was created to combine all the accessions in a single file. SNPs were then extracted and filtered on: QUAL < 30.0, QD < 2.0, FS > 60, MQ < 40, SOR > 3, MQRankSum < –12.5, ReadPosRankSum <–8 and depth <3. The pipeline was implemented in *Snakemake* v5.8.1 [47] and containers were built for all the software using *Singularity* v3.5.3 [48]. We predicted the effect of the polymorphism and declared it “high” in case of a stop gain or loss in the coding sequence, “moderate” in case of a non-synonymous mutation and “low” in case of a synonymous mutation.

## 3. Results

### 3.1. Panel Structure

Since the hybrids shared half of their genome with the corresponding lines, CCI and TCI structures are identical on the PCA (Appendix A): the core collection is spread along the first principal component, starting with a cluster of SLC_INRA mixed with SP_INRA, ending with the remaining SP_INRA. TCT displayed the same pattern, along with additional information on the second axis which opposed the INRAE core collection (SLC_INRAE and SP_INRAE) and the breeding material from Gautier semences (SLC_GS). Only eight accessions were shared between the INRAE core collection and the elite tomato accessions.

### 3.2. Variability of VOCs in the GWAS Panels

A total of 46 VOCs plus SSC and fruit weight remained once we had applied quality filters (Appendix A). Pairwise correlations between the CCI and TCI panels assessed the VOC content stability between lines and F1 hybrids. Overall, 36 VOCs were highly correlated (R > 0.4). SSC and fruit weight also displayed high correlations (0.75 and 0.91, respectively). We then compared phenotypic variances between panels. Except for 11 VOCs and fruit weight, the CCI panel always displayed higher phenotypic variance values than both the TCI and TCT panels, which was expected with the use of a common tester smoothing out differences between F1 hybrids (Appendix A). The inclusion of elite accessions in the TCT panel compared to the TCI panel increased the phenotypic variance and heritabilities of key VOCs, such as all phenylalanine and carotenoid-derived VOCs (Appendix A). Overall, heritabilities were always higher at the homozygous level compared to the F1 hybrid level, except for nine VOCs and fruit weight.

### 3.3. Cluster Analysis and Metabolic Signatures Per Genetic Group

#### 3.3.1. Metabolite Clusters Are Consistent at the Line and F1 Hybrid Levels

Heatmaps were produced for each panel for the 46 VOCs and SSC. Cluster analyses were computed beforehand on metabolites based on the accession profiles. The optimal number of clusters was evaluated at nine, eight and ten for CCI, TCI and TCT, respectively. Whatever the panel considered, metabolite clusters were consistent with those identified in the TCT panel (Appendix A). Clusters were statistically validated as presented in Appendix A on the TCT panel. VOCs belonging to the same metabolic pathway clustered together except a few exceptions that were consistent across panels, and with the study from [49]. The benzenoid-derived BENZA clustered with phenylalanine-derived VOCs in the TCI and TCT panels. The carotenoid-derived metabolic pathway was always divided in two to three clusters, with β-cyclocitral and BIONO isolated together in the TCI and TCT panels. They derive from β-carotene while the other carotenoid-derived VOCs derive from lycopene [50]. The carotenoid-derived BDAM was also isolated from the other carotenoid-derived VOCs in CCI and TCT. Furthermore, lipid-derived VOCs were always divided in two distinct clusters, identical across panels. One cluster mainly contained 7C (seven carbon), 8C and 10C VOCs while the other was enriched in 5C and 6C lipid-derived VOCs associated with ‘green leaf’ aromas.

#### 3.3.2. Genetic Groups Display Characteristic VOC Signatures

In order to identify characteristic features of each genetic group identified in the genetic structure analysis, we conducted a Partial Least Square–Discriminant Analysis (‘PLS-DA’) on each panel (Figure 2) with further statistical validation of VOCs linked to a given genetic group (SP_INRAE, SLC _INRAE and SLC_GS) presented in Appendix A. The genetic groups highlighted in the TCT genetic structure plot could be identified with the same pattern in the TCT PLS-DA (Figure 2a). The SP_INRAE group was enriched in all families of lipid-derived VOCs (Figure 2b), along with SSC and four out of seven carotenoid-derived VOCs. Although to a lesser degree, the SLC_INRAE also showed enrichment in the different families of lipid-derived VOCs. It showed other consistent features with the SP_INRAE, such as high content in branched-chain amino acid-derived VOCs and in the carotenoid-derived 6MHON. Both genetic groups showed depletion in terpenoid-derived VOCs. Overall, there is a continuum in the characteristics of the two groups which is consistent with their domestication history [29]. The major difference in the SLC_INRAE lied in lower SSC, but also in higher content of benzenoid-derived VOCs and the concomitant lower content of phenylalanine-derived VOCs. While Phe-der VOCs are positively correlated to consumer liking, B-der VOCs are on the contrary responsible for off-flavours [8]. On the other hand, the SLC_GS genetic group was quite different. Breeding led to higher content of terpenoid-derived, phenylalanine-derived PHEAC and benzylnitrile and carotenoid-derived BIONO and BDAM VOCs along with higher SSC values. In addition, lipid-derived VOCs with a six carbons chemical structure were selected against. Three carotenoid-derived VOCs had also lower values in this genetic group compared to the overall population. This breeding material displayed its aromatic specificities that resulted in discarding the ‘green leaf’ aroma conferred by the 6C lipid-derived VOCs, while promoting fruity and floral aromas associated with phenylalanine-derived VOCs.

### 3.4. GWAS Analysis

Table 2 summarizes the GWAS results and Appendix A gives the exhaustive list of associations identified for the 46 VOCs, SSC and fruit weight. The CCI, TCI and TCT panels allowed the identification of 205, 138 and 183 associations, respectively. The CI length ranged from 164 bp to 4 Mbp with an average CI length of 0.2 Mbp across panels. The full list of genes harboured in the CIs is presented in Appendix A. The number of genes ranged from one to 73 with an average gene number of 19 across all CIs. Overall, we identified 95, 74 and 118 CGs (Appendix A) based on annotations suggested in the literature for the corresponding metabolic pathways, in addition to 142, 118 and 172 TFs in the CI of the CCI, TCI and TCT panels, respectively. Appendix A lists all possible CGs and TFs from the full list of genes provided by the SL4.0 tomato version of the genome and ITAG.4.0 annotation. A total of 36% of the 287 CGs, and 7% of the 432 TFs presented a cis-eQTL in fruit in [51]. The χ^2^ test on each metabolic pathway and each panel showed significant enrichment of CG presence in the CI of associations from the three panels for SSC and both benzenoid and phenylalanine-derived VOCs. We found enrichment in CGs for the branched-chain amino acid-derived (resp. carotenoid and terpenoid-derived) metabolic pathway for the TCI (resp. TCT) panel. Both CCI and TCT (resp. CCI and TCI) panels showed associations with significant enrichment of CGs from the lipid-derived (resp. sulphur-derived) metabolic pathway. Overall, the MLMM method paired with the definition of narrow CIs allowed us to accurately retrieve CGs.

From the 46 VOCs, SSC and fruit weight measured across panels, 45, 44 and 41 traits displayed at least one association in the CCI, TCI and TCT panels, respectively. The MLMM step-by-step approach allowed the calculation of the percentage of variance explained (‘PVE’) for the first association of each trait. The CCI, TCI and TCT panels had 76%, 90% and 90% of their first associations with PVE > 20%, respectively. These first associations, accounting for 24% of the 526 total associations identified, carried 21% of the total number of CGs.

In order to identify promising associations among those presented in Appendix A, we looked for overlapping associations within each metabolic pathway. Then, we focused on associations that overlapped between lines (CCI) and F1 hybrids (TCI or TCT), or F1 hybrid specific (TCI or TCT). We thus identified 30%, 16% and 22% of associations solely found in CCI, TCI and TCT, respectively. Among them, 17, four and 10 associations were shared by two or more traits within the panel and the metabolic pathway for CCI, TCI and TCT, respectively (association redundancy is specified in Appendix A ‘association.redundancy’ column). Our aim was to identify either associations F1 hybrid specific or associations shared between F1 hybrids and lines panels. As a result, nine and five associations overlapped between CCI and TCI, and CCI and TCT, respectively. In addition, 20 associations overlapped between TCI and TCT and six associations were consistent between the three panels.

Overall, we retrieved four previously cloned genes among our associations. The *smoky* gene (Solyc09g089585) cloned by [52] was identified in the CI of identical associations for three benzenoid-derived VOCs in the three different panels. The glycosyltransferase (Solyc04g064490) highlighted by [22] for the phenylalanine-derived PHEAC and PHENE mapped in the CI of PHENE in the TCT panel. We also identified the alcohol dehydrogenase *ADH2* (Solyc09g025210) cloned by [53] involved in the lipid-derived metabolic pathway in the CI of the lipid-derived (E,E)-2,4-heptadienal of the TCT panel. Finally, we retrieved *Lin5* (Solyc09g010080) [54] involved in the accumulation of sugars in fruits in the three panels for SSC.

### 3.5. Candidate Genes for F1 Hybrid Quality

Apart from cloned genes, two, five and 14 associations found in the CCI, TCI and TCT panel, respectively, carried 17 different CGs previously suggested in [20] (Appendix A). Ten CGs belonged to associations for lipid-derived VOCs, three CGs for benzenoid-derived VOCs and three CGs for phenylalanine-derived VOCs. Six CGs will be further detailed as they belong to the most promising regions where we identified novel CGs. To complete our goals for both fundamental and applied research, we first highlighted the most promising CGs underlying key VOCs, and then we defined candidate regions with potential for multi-VOCs improvement. Whatever the purpose, we filtered our associations based on key criteria, all aimed at improving tomato taste in F1 hybrids: (1) regions carrying associations with *p*-values < 10^−6^ for the subset of VOCs impacting tomato aroma; (2) markers in association with balanced Minor Allele Frequency (‘MAF’) (>0.2) to ensure that both allelic classes were well represented, before estimating FTMV allelic effect on the VOC content; (3) regions displaying overlaps between VOCs belonging to the same metabolic pathway, and with consistent effect of FTMV allelic class on closely related VOCs.

All CGs identified in the CI of our associations are presented in Appendix A. Details about the associations carrying these CGs can be found in Appendix A. We selected a subset presented in Table 3 and Figure 3 that gathered the most promising candidates to achieve flavour improvement in F1 hybrids, based on the strategy we defined. We checked the polymorphisms among these CGs in the 47 resequenced lines of the CCI panel. Out of the 33 CGs we suggested in Table 3, 16 presented at least several low (synonymous mutation) to moderate (non-synonymous mutation) effect polymorphisms in their coding region, and six presented high effect (stop gain or loss) polymorphism. Details about these polymorphisms can be found in Appendix A.

#### 3.5.1. Lipid-Derived VOCs

Many lipid-derived VOCs are supposed to impact tomato aroma as reviewed in Table 1. We thus suggested CGs carried in “CG.L.1” and “CG.L.3” to impact L-der VOC content in tomatoes. “CG.L.1” encompassed only one association on chromosome 1 in the TCT panel for 1P3ON, which CI carried 13 genes. Among them, we identified five lipoxygenase genes previously suggested in [20] review. These gene functions are known to participate in various plant mechanisms and biosynthesis processes, among which the metabolism of linoleic and linolenic acids into fatty acid hydroperoxides, further catabolized in VOCs [55]. We identified high effect polymorphism in two lipoxygenase genes (Solyc01g099170 and Solyc01g099210). The authors of [56] associated 1P3ON with fresh and sweet notes, and suggested the accumulation of this VOC in tomato fruit to improve consumer acceptance. While “CG.L.1” could specifically impact 1P3ON accumulation in fruit, “CG.L.3” encompassed the regulation of seven VOCs on chromosome 3. This region is specific to hybrid panels, with a common marker associated with seven different VOCs, and five associations being common between TCI and TCT. Four out of these seven VOCs impact tomato aroma. All associations carried a lipase (Solyc03g083370) suggested in [20], which harboured ten non-synonymous mutations in its coding region. Catabolism of linolenic or linoleic acids produces structurally different lipid-derived VOCs, with five and six carbons VOCs with the first, and six to ten carbons VOCs with the latter [20]. Since we found associations for all these different VOCs, we expect that this gene might regulate the whole lipid-derived metabolic pathway by converting acylglycerides in both linolenic and linoleic acids.

#### 3.5.2. Phenylalanine and Benzenoid-Derived VOCs

Regulating phenylalanine and benzenoid-derived VOCs is important for flavour improvement. Although they both derive from the amino-acid phenylalanine, they confer aromas with opposite impact on consumer liking: all the phenylalanine-derived VOCs we quantified are positively correlated to consumer liking with sweet and floral aromas, while most benzenoid-derived VOCs are negatively correlated to consumer liking with ‘smoky’ and ‘medicinal’ aromas. Identifying key regulators of their synthesis is thus of utmost interest. We suggested “CG.B.Phe.3” that showed overlapping associations for EUGEN in TCI and PHEAC in TCT on chromosome 3, both carrying a glycosyltransferase (Solyc03g118120) that showed a cis-regulation pattern, which might result from at least one of the four non-synonymous mutation in its coding region [22,52]. “CG.B.4” carried overlapping associations on chromosome 4 in TCT for the benzenoid-derived VOCs EUGEN and GUAIA. It contained the key regulator DAHP synthase 2 precursor (Solyc04g074480) which showed a cis-regulation pattern and one non-synonymous mutation in its coding region. Solyc04g074480 provides the first step to synthesize phenylalanine [57]. “CG.B.8” was defined by an association in TCT on chromosome 8 for the benzenoid-derived BENZA. Its CI carried seven glycosyltransferase genes and 11 TFs. Four of the glycosyltransferase genes exhibited cis-regulation patterns and carried non-synonymous mutations and three of them also carried a high effect polymorphism. The authors of [20] also highlighted the UDP-xylose phenolic glycosyltransferase (Solyc08g006330) as a potential CG. With “CG.Phe.9”, we targeted the genetic control of phenylalanine-derived VOCs with overlapping associations on chromosome 9 between CCI and TCI for PHENE, PHEAC and 1N2PHENE. This region carried a glycosyltransferase (Solyc09g011090) harbouring five non-synonymous mutations. Lastly, for “CG.Phe.12” defined by an association in TCT on chromosome 12 for the phenylalanine-derived PHEAC we identified a glycosyltransferase (Solyc12g010200) with two non-synonymous mutations.

### 3.6. Breeding for F1 Hybrid Quality

Table 4 and Figure 3 highlight seven of the most interesting regions for breeding F1 hybrids with improved aromatic quality. For each region, we specified whether the homozygous or heterozygous genotype would be preferable to achieve this flavour improvement. We also described the breeding potential of the CG regions presented in Table 3.

Based on the aroma perception of VOCs, a better aroma quality should be obtained when the markers in association in regions “Breed.2”, “Breed.3”, “Breed.9” and “Breed.11”, and also those of all CG regions from Table 3, except “CG.B.8” are heterozygous. On the contrary homozygosity of the FTMV alleles for the markers associated with regions “Breed.4”, “Breed.6”, “Breed.12” and “CG.B.8” would confer aromatic advantage. “Breed.3”—that encompassed “CG.L.3” on chromosome 3— and “Breed.6” on chromosome 6 are two regions providing a consistent and overall regulation of lipid-derived VOCs, with associations only found in hybrid panels. Similarly, “Breed.12” also carried hybrid specific associations for all terpenoid-derived VOCs on chromosome 12. On chromosome 4, “CG.B.4” could contribute to decreasing benzenoid-derived VOC content with associations found in TCI for two VOCs. Targeted breeding for single key VOCs could be achieved with “CG.L.1” on chromosome 1, “CG.B.8” on chromosome 8 and “CG.Phe.12” on chromosome 12 to increase 1P3ON, BENZA and PHEAC, respectively. Specialized regulation of metabolic pathways could be achieved through “CG.B.Phe.3” on chromosome 3 and “Breed.4” on chromosome 4: given the overlapping associations we encountered in these two regions, breeders could drive the metabolism of the common phenylalanine precursor to the phenylalanine-derived VOCs, at the expense of the benzenoid-derived VOCs. “Breed.11” carried associations TCT specific for all terpenoid-derived VOCs, with specialized regulation of VOCs according to their closeness in the cluster analysis (Appendix A). One could thus increase LIN and α-terpineol while decreasing (E)-linalool oxide and (Z)-linalool oxide. “Breed.2” regulated VOCs belonging to different metabolic pathways, and would serve as a lever to decrease the benzenoid-derived EUGEN while increasing the branched-chain amino acid-derived 1N3MBUT and lipid-derived HEPTAL if the heterozygous allelic combination were to be selected at the markers in association. Finally, “Breed.9” gave new insight into the value of the region carrying *Lin5* where cherry type tomato alleles increased the amount of sugar. Adjacent to *Lin5*, we identified associations highlighted in “CG.Phe.9” for three phenylalanine-derived VOCs contributing to consumer liking. Choosing heterozygosity over FTMV homozygosity could thus increase SSC along with key VOCs in this region. However, we found associations for fruit weight within “Breed.2” and “Breed.3”, and near “Breed.9” and “CG.B.4”. The cloned gene *fw2.2* is located two genes upstream “Breed.2”. All of these fruit weight associations showed higher fruit weight value for FTMV homozygous genotype.

## 4. Discussion

Most of the efforts in tomato flavour have been directed to understand the genetic control of VOC accumulation in homozygous lines [20]. However, little is known on how the QTLs behave in the F1 hybrids where many of the flavour alleles could be heterozygous. To the best of our knowledge, this study is the first to address this issue by studying a GWAS panel at the homozygous and heterozygous levels, along with another expanded panel comprising additional elite cherry tomato F1 hybrids. With these three panels, we characterized the phenotypic diversity of 46 VOCs among a core collection of small fruited accessions at the line (CCI) and F1 hybrid (TCI) levels, in addition with considering the genetic resource and aromatic improvements that are already available in elite plant material (TCT). We identified numerous genomic regions that regulate key VOCs, with less than 10% overlap between line and F1 hybrid panels. We developed a strategy for F1 hybrid flavour improvement, broken down in two points: (i) Mining for novel CGs revealed in F1 panels to better understand VOC genetic control at the hybrid level. (ii) Achieving flavour improvement by suggesting candidate regions carrying overlapping associations in F1 hybrid panels for key VOCs, and consistent effect of the common tester FTMV genotype on closely related VOCs.

### 4.1. Targeting Appropriate Plant Material to Better Elucidate VOC Genetic Control

As previous studies highlight [8], the complexity of flavour traits and the lack of suitable markers led to fewer allelic diversity remaining in modern cultivars. This led to loss of flavour, although [58] provided evidence of a considerable increase in allelic diversity in breeding material since the 1970s thanks to numerous introgressions performed from wild related species, mainly for introgression of disease resistance genes and to the commercialization of cherry tomato types due to their interesting flavour. In order to avoid the genetic load carried by wild species, we used small cultivated tomato accessions and SP species (closest wild ancestor of the tomato). We compared the diversity in aroma composition of these plant materials at the line and F1 hybrid levels to better understand flavour inheritance and thus suggest a strategy to breed tastier tomatoes. Taking together the genetic structure analysis (Appendix A) and the PLS-DA on VOC content (Figure 2), we found consistent overlapping patterns of hybrids belonging to different genetic groups in both genetic structure and PLS-DA, suggesting that VOC specificities resulted from allelic specificities, thus ruling out an overall environmental control of VOC abundancy for this study. Furthermore, the correlations calculated between the line (CCI) and hybrid (TCI) panels for similar VOCs were high, with 80% of VOCs displaying R > 0.4 between the two panels. These results suggested a global additivity pattern for most VOC genetic control. Heritabilities were high for most VOC content, although they were generally lower in hybrid panels. Accordingly, we found more associations in the CCI panel compared to the F1 panels, although the TCT panel gathered more phenotypic variability and genetic diversity for some key VOCs leading to more associations despite lower heritabilities. VOCs with high heritabilities offer better response to selection by targeting the genetic loci they rely on. As appeared in PLS-DA when comparing the core collection at the line (CCI) and F1 hybrid (TCI) levels, the hybrid panels still expressed their phenotypic specificities although they shared half of their genotypic information with one another: VOCs discriminating the genetic groups SP and SLC were indeed close to identical (Appendix A) between the CCI and TCI PLS-DA results. Similarly, the cluster analysis on the three panels showed consistent to similar results indicating that being at the line or hybrid level, VOC regulation showed the same patterns, which further supported the additivity hypothesis. Moreover, our clusters matched reviews published on VOC metabolic pathways in tomato fruits [6,8], and the VOC cluster analysis performed on a biparental population derived from a cross between an SP and an SL accession in [49]. Considering the TCT panel that harboured 44 additional F1 hybrids produced from elite cherry type tomatoes and the common parent FTMV, there is evidence of the emphasis breeders put on creating cherry type tomato varieties with improved aroma composition (Appendix A), notably they showed enrichment of the phenylalanine-derived PHEAC, the carotenoid-derived BIONO and BDAM and sugar content, and concomitant reduction of four out of five branched-chain amino acid-derived VOCs and 6C lipid-derived VOCs (Appendix A). These compounds are known to impact tomato aroma and were selected, or selected against by breeders without prior quantification. The elite plant material showed enrichment in terpenoid-derived VOCs, which although rarely studied, could also be key VOCs for consumer liking as highlighted in [6]. VOC signatures highlighted in the PLS-DA gave insight into each genetic group specificities, although the heatmap presented in Appendix A showed that phenotypic diversity could be found back in each genetic group, thus allowing breeders to target donor accessions to improve their breeding plant material.

### 4.2. Strategy for F1 Hybrids Quality Improvement

As stated in [8], flavour is a highly polygenic trait that requires a robust strategy for quality improvement to be achieved. We know the metabolic pathways involved in VOC synthesis, some of the underlying gene functions, and even more importantly we know a subset of approximately 30 VOCs that impact tomato aroma. Based on this information that constitutes the first step of a strategy to improve tomato aroma, we herein tackled the next central issue for tomato breeders: how does heterozygosity encountered in commercialized F1 hybrids affect VOC abundancy, and which allelic combinations would be preferable to achieve sensible flavour improvement with so many target traits.

Given the common regulation of volatiles belonging to the same metabolic pathway as found in the cluster analysis, an overall regulation of key metabolic pathways could be achieved by selecting overlapping associations for VOCs clustering with one another. Since we cannot estimate yet how many favourable alleles are necessary to make significant flavour differences in the breeding material, we proposed the most important breeding regions encompassing these overlaps to achieve multi-VOC regulation. We also indicated regions where promising CGs could be found based on the same criteria. The next and final criteria was selecting among these regions those that displayed a consistent effect of FTMV homozygous genotype across VOCs belonging to the same metabolic pathway and on VOCs with similar impact on consumer liking in order to (i) control and understand the taste differences that could be accomplished and (ii) increase VOCs with positive impact on consumer liking while simultaneously decreasing those leading to off-flavours.

For this strategy to work on F1 hybrids, we highlighted regions with F1 specific associations, or overlapping between F1 hybrid (TCI TCT) and line (CCI) panels. With only nine and five associations commonly found between CCI and TCI and CCI and TCT, respectively, and six associations shared between the three panels, we underlined the utmost importance of considering associations fit to design F1 hybrid varieties instead of considering QTLs identified only on homozygous plant material. We revealed novel regions where phenotypic means for a given volatile are more contrasted when we compare the homozygous and heterozygous locus versions versus the two homozygous genotypes at the locus. Although the environmental effect causing lack of overlaps between line and hybrid panels cannot be ruled out, the high correlations calculated between CCI and TCI are evidence that novel associations in F1 hybrids did not result solely from an environmental effect. Additionally, we defined stringent CIs around associated markers by accounting for both population structure and kinship when calculating local LD between neighbouring markers, and we called “overlap” CI boundaries separated by less than 0.1 Mbp. Those parameters led to few overlaps. With less stringent parameters, more overlaps would be detected. Apart from the four cloned genes that we found back in hybrid panel associations and 6% of hybrid specific associations for VOCs carrying CGs previously suggested in [20], we identified novel associations for VOCs. Nonetheless, the χ^2^ test we performed on each metabolic pathway and each panel separately mostly showed significant enrichment of CG presence in the CI of our associations. This underlined that the MLMM method paired with the definition of narrow CIs allowed us to retrieve CGs accurately.

As part of our strategy, we also gave insight into the best allelic combinations for breeding by giving the relative position of the heterozygous class compared to FTMV homozygous genotype for a given trait-marker association. For each breeding region we selected, and even more in the elite plant material, we found associated markers with balanced MAF, which highlighted that many regions in small fruited accessions showed the allelic version of the common parent FTMV entirely lacking flavour. This finding raised awareness about the improvement that could be achieved given the number of novel levers for key VOCs we herein proposed, and that remained dormant up until now.

Despite growing interest in organoleptic quality in commercialized hybrids, maintaining yield and other agronomical traits remains an important concern. We therefore assessed fruit weight in our experimental analyses. We chose a SL common tester to: (i) increase heterozygosity occurrence; (ii) study its impact on VOC content; and (iii) address both cherry type and big fruited tomato flavour improvement in light of fruit weight associations among the other VOC quality trait associations we identified. We found 11 hybrid specific associations for fruit weight, three of them mapping within 1 Mb of the cloned genes *fw2.2*, *lc* and *fw11.2*, respectively. Only one fruit weight association, close to *Lin5*, overlapped between the line and hybrid panels. We indicated regions where flavour improvement could be accomplished, though at the expense of fruit weight. Thus, “Breed.2” with the different metabolic pathways it regulated and “Breed.3”, “Breed.9” and “CG.B.4” with global regulation of the lipid, phenylalanine and benzenoid-derived VOCs, respectively, were rather aimed at cherry type tomato breeding than big fruit type. Otherwise, we showed that improved flavour could be achieved without trade-off with yield for other breeding regions. Thus, our results can be of interest to breeders seeking to create parental lines with better aroma composition potential while being less likely to have an effect on fruit weight.

## Figures and Tables

**Figure 1 genes-12-01443-f001:**
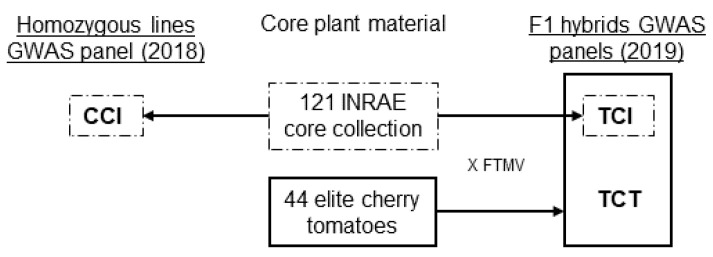
Plant materials in the three GWAS panels. We built the panels starting from the homozygous core plant materials. We studied the INRAE core collection in 2018 at the line level (CCI). We then studied a test cross in 2019 (TCT) obtained from the overall core plant materials crossed with the same big fruited tester FTMV. To compare the core collection at the line and F1 hybrid levels, we extracted the test cross carrying only the core collection F1 hybrids (TCI) from the TCT panel and studied this third panel independently.

**Figure 2 genes-12-01443-f002:**
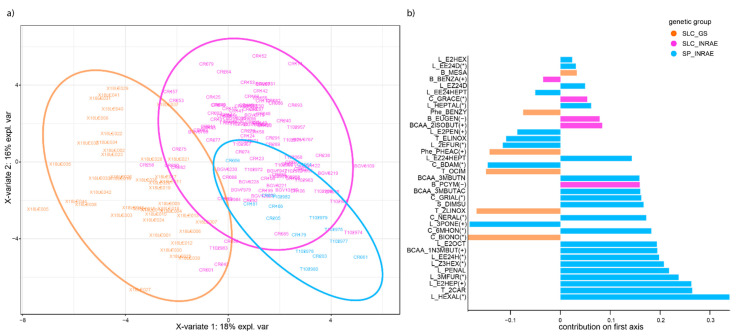
PLS-DA on VOC and sugar content in the three genetic groups of the TCT panel. (**a**) F1 hybrid plot. Each colour corresponds to a genetic group in the TCT panel. (**b**) VOCs maximizing the discrimination between the genetic groups on the first axis of the F1 hybrid plot. The colour of the contribution highlights the genetic group which shows enrichment in the corresponding VOC. The direction of the contribution depends on the position of the genetic group. VOC impact on tomato aroma is indicated at the end of the VOC name as presented in Table 1. +: positive correlation to consumer liking, −: negative correlation to consumer liking, *: impacts flavour perception.

**Figure 3 genes-12-01443-f003:**
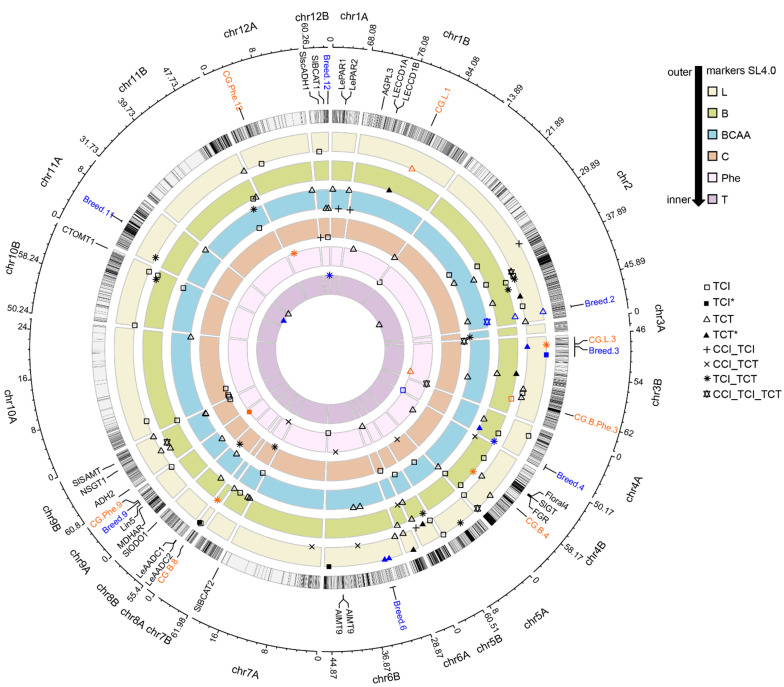
Circos plot of associations found for VOCs impacting tomato aroma. The outmost lane gives the marker density on the 12 chromosome arms (centromeric region are not displayed). Each successive lane represents a specific metabolic pathway. Association shape depends on the panel in which they were found. An asterisk (*) indicates overlapping associations from the same panel, with only one association plotted in case of overlap. Associations with *p* value < 10^−6^ are plotted on the outer end of the lane, otherwise they are plotted on the inner end of the lane. We displayed positions of cloned genes impacting tomato aroma, along with the breeding (blue) and candidate gene (orange) regions we selected on top of the marker lane.

**Table 1 genes-12-01443-t001:** Important VOCs for consumer liking among the 46 VOCs, SSC and fruit weight (adapted from [8]).

Full Name	VOC ID	Met. Path	Aroma. Impact	Aroma. Descriptors
Benzaldehyde	BENZA	B	+	Almond, burnt sugar, peaches, fruity
Eugenol	EUGEN	B	−	Sweet, spicy, clove, woody, pharmaceutical
Guaiacol	GUAIA	B	*	Phenolic, smoke, spice, vanilla, woody
p-cymene	PCYM	B	*	Solvent, gasoline, citrus
Salicylaldehyde	SALI	B	−	NA
1-nitro-3-methylbutane	1N3MBUT	BCAA	+	NA
2-isobutylthiazole	2ISOBUT	BCAA	+	Tomato, leafy, green, pungent, medicinal
3-methylbutanal	3MBUT	BCAA	*	Ethereal, aldehydic, chocolate, peach, fatty, malt
β-damascenone	BDAM	C	*	Apple, rose, honey, tobacco, sweet
β-ionone	BIONO	C	*	Floral, sweet
Geranylacetone	GRACE	C	*	Sweet, floral, estery, citrus
Geranial	GRIAL	C	*	Sharp, lemon, sweet
Neral	NERAL	C	*	Lemon
6-methyl-5-hepten-2-one	6MHON	C	+	Fruity, floral, sweet
(E)-2-heptenal	E2HEP	L	+	Dried fruits
(E)-2-pentenal	E2PEN	L	+	NA
(E,E)-2,4-decadienal	EE24D	L	*	Earthy, musty
(E,E)-2,4-hexadienal	EE24H	L	*	Green
Heptanal	HEPTAL	L	*	Fat, citrus, rancid
Hexanal	HEXAL	L	*	Grass, tallow, fat
1-penten-3-one	1P3ON	L	+	Fruity, floral, green, fresh, sweet
2-ethylfuran	2EFUR	L	*	Rum, coffee, chocolate
3-methylfuran	3MFUR	L	*	NA
3-pentanone	3PONE	L	+	NA
(Z)-3-hexenal	Z3HEX	L	*	Leafy, green, grass, tomato
Phenylacetaldehyde	PHEAC	Phe	+	Hawthorne, honey, sweet
2-phenylethanol	PHENE	Phe	+	Honey, spice, rose, lilac
1-nitro-2-phenylethane	1N2PHENE	Phe	+	Flower, spice
Linalool	LIN	T	*	Citrusy, fruity, sweet taste

VOCs impacting flavour perception are marked with a (*) in column “Aroma.impact” when their odour threshold or abundancy is at stake (reviewed in [14,15,16]). VOCs with positive (resp. negative) significant correlation to consumer liking are marked with a (+) (resp. (−)) [8] (adapted from [8]). VOCs abbreviations are specified in “VOC ID” B: benzenoid, BCAA: branched-chain amino acid, C: carotenoid, Phe: phenylalanine, T: terpenoid, NA: not available.

**Table 2 genes-12-01443-t002:** GWAS summary for SSC and VOCs impacting tomato aroma.

Met. ID	Met. Path	Aroma	Mean		h2			Nb. Assoc		
			FTMV	TCT	CCI	TCI	TCT	CCI	TCI	TCT
BENZA	B	+	1.91	1.21	0.63	0.36	0.32	1	1	4
EUGEN	B	−	1.33	1.18	0.63	0.75	0.76	8	9	11
GUAIA	B	*	1.60	0.68	0.86	0.76	0.74	7	3	3
PCYM	B	*	0.82	1.07	0.72	0.48	0.44	4	6	3
SALI	B	−	1.44	0.98	0.77	0.67	0.66	11	2	6
1N3MBUT	BCAA	+	1.49	1.55	0.55	0.30	0.37	2	5	10
2ISOBUT	BCAA	+	1.36	1.12	0.68	0.43	0.45	1	2	1
3MBUT	BCAA	*	1.94	1.22	0.51	0.16	0.08	4	3	10
6MHON	C	+	1.04	1.15	0.62	0.32	0.47	4	1	0
BDAM	C	*	0.64	0.99	0.75	0.53	0.55	0	6	2
BIONO	C	*	0.91	1.18	0.41	0.18	0.35	8	0	2
GRACE	C	*	1.15	1.48	0.40	0.34	0.30	4	1	0
GRIAL	C	*	1.02	1.16	0.60	0.32	0.43	6	1	0
NERAL	C	*	1.14	1.13	0.60	0.22	0.36	3	0	0
1P3ON	L	+	0.90	1.03	0.77	0.52	0.55	4	4	7
2EFUR	L	*	0.99	1.04	0.32	0.38	0.46	1	5	0
3MFUR	L	*	0.84	1.22	0.40	0.25	0.36	3	4	1
3PONE	L	+	1.14	1.17	0.49	0.28	0.22	1	3	3
E2HEP	L	+	0.84	1.05	0.33	0.19	0.36	1	1	3
E2PEN	L	+	0.88	1.12	0.65	0.45	0.56	2	3	5
EE24D	L	*	0.86	1.13	0.61	0.35	0.33	4	0	0
EE24H	L	*	1.13	0.94	0.65	0.41	0.51	0	2	3
EZ24D	L	*	0.77	1.02	0.65	0.25	0.31	3	0	4
HEPTAL	L	*	0.97	1.10	0.30	0.19	0.02	5	1	5
HEXAL	L	*	0.92	0.96	0.51	0.25	0.56	3	1	3
Z3HEX	L	*	1.04	0.99	0.53	0.29	0.41	3	2	4
1N2PHENE	Phe	+	0.97	1.31	0.71	0.48	0.52	9	4	0
BENZY	Phe	*	1.99	1.12	0.85	0.61	0.58	9	4	0
PHEAC	Phe	+	1.97	1.22	0.71	0.35	0.33	4	3	11
PHENE	Phe	+	1.40	1.29	0.71	0.55	0.59	4	1	2
ELINOX	T	*	0.66	1.12	0.80	0.62	0.59	5	2	4
LIN	T	*	0.94	0.99	0.03	0.32	0.30	1	1	4
ZLINOX	T	*	0.62	1.13	0.81	0.67	0.65	3	5	4
SSC	SUGAR	+	4.70	5.88	0.82	0.77	0.73	6	3	2

We indicated the mean values of the common tester FTMV and the mean values in the overall TCT panel, both measured alongside in 2019. Heritability (‘h2’) is indicated along with the number of associations identified per panel (‘nb.assoc’). VOCs impacting flavour perception are marked with a (*) in column “Aroma.impact” when their odour threshold or abundancy is at stake (reviewed in [56,57,58]). VOCs with positive (resp. negative) significant correlation to consumer liking are marked with a (+) (resp. (−)) [8] (adapted from [8]). VOCs abbreviations are specified in “VOC ID” B: ben-zenoid, BCAA: branched-chain amino acid, C: carotenoid, Phe: phenylalanine, T: terpenoid.

**Table 3 genes-12-01443-t003:** Candidate genes and transcription factors identified in regions defined over associations for VOCs involved in tomato aroma.

ID	VOCs	Solyc. ID	Function	Exp	eQTL	Low	Mod	High
CG.L.1	1P3ON	Solyc01g099160	Lipoxygenase	+	−	7	19	0
CG.L.1	1P3ON	Solyc01g099170	Lipoxygenase	+	−	1	5	1
CG.L.1	1P3ON	Solyc01g099190	Lipoxygenase B	+	−	6	7	0
CG.L.1	1P3ON	Solyc01g099200	Lipoxygenase	+	−	3	2	0
CG.L.1	1P3ON	Solyc01g099210	Lipoxygenase	+	−	3	5	1
CG.L.3	3PONE-E2HEX-E2PEN-EE24H-EE24HEPT-EZ24HEPT-Z3HEX	Solyc03g083370	GDSL esterase/lipase	+	−	8	10	0
CG.B.Phe.3	EUGEN-PHEAC	Solyc03g118120	Glycosyltransferase	+	+	3	4	0
CG.B.Phe.3	EUGEN-PHEAC	Solyc03g118190	Transcription factor	+	+	0	0	0
CG.B.Phe.3	EUGEN-PHEAC	Solyc03g118230	Transcription factor	+	−	0	0	0
CG.B.Phe.3	EUGEN-PHEAC	Solyc03g118310	Transcription factor	+	−	0	0	0
CG.B.4	EUGEN-GUAIA	Solyc04g074480	DAHP synthase 2	+	+	2	1	0
CG.B.8	BENZA	Solyc08g006110	Transcription factor	+	−	0	0	0
CG.B.8	BENZA	Solyc08g006190	Transcription factor	+	−	0	0	0
CG.B.8	BENZA	Solyc08g006200	Transcription factor	+	−	0	0	0
CG.B.8	BENZA	Solyc08g006210	Transcription factor	+	−	0	0	0
CG.B.8	BENZA	Solyc08g006220	Transcription factor	+	+	0	0	0
CG.B.8	BENZA	Solyc08g006230	Transcription factor	+	−	0	0	0
CG.B.8	BENZA	Solyc08g006240	Transcription factor	+	−	0	0	0
CG.B.8	BENZA	Solyc08g006270	Transcription factor	+	−	0	0	0
CG.B.8	BENZA	Solyc08g006280	Transcription factor	+	−	0	0	0
CG.B.8	BENZA	Solyc08g006320	Transcription factor	+	−	0	0	0
CG.B.8	BENZA	Solyc08g006330	Glycosyltransferase	+	+	14	14	1
CG.B.8	BENZA	Solyc08g006350	Glycosyltransferase	+	+	10	15	1
CG.B.8	BENZA	Solyc08g006360	Glycosyltransferase	+	+	7	4	0
CG.B.8	BENZA	Solyc08g006370	Glycosyltransferase	+	+	1	9	1
CG.B.8	BENZA	Solyc08g006390	Glycosyltransferase	−	−	2	6	1
CG.B.8	BENZA	Solyc08g006400	Glycosyltransferase	−	−	0	0	0
CG.B.8	BENZA	Solyc08g006410	Glycosyltransferase	+	−	5	4	0
CG.B.8	BENZA	Solyc08g006483	Transcription factor	−	−	0	0	0
CG.Phe.9	PHEAC-1N2PHENE-PHENE	Solyc09g011090	Glycosyltransferase	−	−	4	5	0
CG.Phe.9	PHEAC-1N2PHENE-PHENE	Solyc09g011110	Transcription factor	+	−	0	0	0
CG.Phe.12	PHEAC	Solyc12g010170	Transcription factor	+	−	0	0	0
CG.Phe.12	PHEAC	Solyc12g010200	Hexosyltransferase	+	−	3	2	0

The function of each gene is given according to the ITAG4.0 annotation. Associations carrying candidate gene regions can be retrieved in Appendix A column ‘association.status’ along with the FTMV genotype at the corresponding markers. ‘exp’ represents expression data based on the fruit transcriptome analysis from [41]. ‘eQTL’ indicates genes for which [51] found a cis regulation eQTL. The number of polymorphisms detected in the coding region of the genes are indicated according to their effect (High: stop codon; Mod: non-synonymous mutation; Low: synonymous mutation). GDSL: named after the conserved motif ‘Gly-Asp-Ser-Leu’, DAHP: 3-Deoxy-D-arabino-heptulosonic acid 7-phosphate,+: gene expressed in tomato fruit/gene with a cis-regulation eQTL, −: gene not expressed in tomato fruit / gene with no known cis-regulation eQTL.

**Table 4 genes-12-01443-t004:** Candidate regions for flavour improvement.

ID	Met. Path	Flav. Geno	Position		FTMV.Effect	
			Chr	Start–End	Increases	Decreases
Breed.2	B-BCAA-L	F/C	2	50.05–50.46	EUGEN(−)	1N3MBUT(+) HEPTAL(*)
Breed.3	L	F/C	3	46.44–50.25	/	HEPTAL(*) 1P3ON(+) 3PONE(+) E2PEN(+) EE24H(*) Z3HEX(*) E2HEP(+) 2EFUR(*)
Breed.4	Phe-B	F/F	4	5.79–7.23	1N2PHENE(+)	EUGEN(−) GUAIA (*)
Breed.6	L	F/F	6	31.61–33.17	E2PEN(+) 1P3ON(+) 3PONE(+) EE24H (*)	/
Breed.9	Phe	F/C	9	3.51–4.90	/	1N2PHENE(+) PHENE(+)
Breed.11	T	F/C	11	4.58–4.97	/	LIN(*)
Breed.12	T	F/F	12	63.98–64.08	LIN(*)	/

The intervals of the regions are given in Mbp according to the SL4.0 version of the tomato genome. We indicate the effect of FTMV homozygous genotype only for VOCs impacting tomato flavour as presented in Table 1. The full list of VOCs impacted in the region can be found in Appendix A column ‘association.status’, along with FTMV genotype at the corresponding markers. According to the bibliography, we suggest the genotype to achieve better flavour (‘flav.geno’) at the markers carried in the regions as ‘F/F’ for the homozygous FTMV genotype or F/C for the heterozygous genotype. +: positive correlation to consumer liking, −: negative correlation to consumer liking, *: affects flavour perception.

## Data Availability

The CCI and TCI phenotypes are available in Appendix A, and the FTMV phenotype in Appendix A. The complementary TCT phenotypes are available on demand. The common tester FTMV and the CCI accessions raw genotypic data are available in Appendix A.

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
