# Peer review of "Breeding Tomato Hybrids for Flavour: Comparison of GWAS Results Obtained on Lines and F1 Hybrids"

_genes, 2021, doi:10.3390/genes12091443_

Round 1
Reviewer 1 Report
Bineau et al presented a very interesting study of comparing GWAS among hybrid and parental lines. However, I have some concerns that I need it should be addressed in the manuscript.
- In most places author used short form of different terminology. I think they used full forms just to avoid confusion.
- How authors have calculated genetic correlation .Need to mention in the method section
- Authors have identified candidates. However they should mention about the SNPS present in those genes and how it is different from parental strain
- Result and discussion sections are too long can be shortened and precise
- Authors should mention the reason of doing test cross in the text.
- Hybrid tomatoes longevity should also be check and compared from that of parental species
Author Response
We would like first to thank both reviewers for their positive review and suggestions.
Below we answer point by point to their suggestions and all modifications can be tacked in the revised manuscript.
- In most places author used short form of different terminology. I think they used full forms just to avoid confusion.
We agree, but removing all abbreviations could be also very difficult to follow, so we made specific modifications:
We debated over keeping the full description of the different panels when we refer to one specifically, but chose instead abbreviations to ease the reading of figures tables and manuscript. We explain our abbreviations in figure 1 and in the Materials and Methods “Plant material” subsection.
For volatiles, we modified our abbreviations so that:
- metabolic pathways are abbreviated in figures and tables but not in the text
- volatiles with a known impact on consumer liking are abbreviated in the text as presented in the new “Table 1”, and those with no known impact on consumer liking are cited in full in the text, but abbreviated in figures. Their abbreviation can be found back in Table S5
- How authors have calculated genetic correlation. Need to mention in the method section
Line 220-221 we added: “Structure and Kinship matrices were reevaluated for each trait in the GWAS analyses with the R package FactomineR and the function emma.kinship (Identity By State method)”
3. Authors have identified candidates. However they should mention about the SNPS present in those genes and how it is different from parental strain
We looked for polymorphisms in the coding sequence of the candidate genes presented in Table 3. We identified polymorphisms based on the method described in the subsection ‘2.8. Putative effect of polymorphisms in candidate genes’. We added a new supplementary table (Table S8) to specify the position of such polymorphisms, the reference allele of Heinz_1706, and the alternate allele encountered among the 47 resequenced lines. We do not state the tester FTMV allele for the polymorphisms we identified, as we do not have this information.
- Result and discussion sections are too long can be shortened and precise
We removed several sentences in Results and Discussion (see track changes underlined)
5. Authors should mention the reason of doing test cross in the text.
- We added in the introduction line 100-101: “a third panel derived from the test cross, carrying only the 121 hybrids from the core collection to compare VOC inheritance at the line and hybrid levels (Figure 1).”
- In the caption of Figure 1: “To compare the core collection at the line and F1 hybrid levels, we extracted the test cross carrying only the core collection F1 hybrids (TCI) from the TCT panel and studied this third panel independently.”
- In the discussion Lines 572-574 “We compared the diversity in aroma composition of these plant materials at the line and F1 hybrid levels to better understand flavour inheritance and thus suggest a strategy to breed tastier tomatoes.”
6. Hybrid tomatoes longevity should also be check and compared from that of parental species
We didn’t find this information relevant for this article, as we focus only on flavour compounds. Several papers already detail shelf life in hybrids using major genes such as rin, nor and alc.
Reviewer 2 Report
This is a very complex study about VOCs of tomatoes using a wide variety of methods and combines genome wide association studies based on former studies about tomatoes (i.e. QTL). Studies on the influence of heterozygosity in F1 hybrids of tomatoes was - according to the authors- done for the first time. Most interestingly the authors fount additivity of most VOCs, but they could specify whether the homozygous or heterozygous genotype would be preferable to achieve a better flavour. Moreover, they present markers responsible for this important breeding step.
I only have a minor point, write all scientific plant names in italic (i.e. line 92) and following.
Author Response
Reviewer 2. I only have a minor point, write all scientific plant names in italic (i.e. line 92) and following.
Response : This was changed lines 97, 99 and 693